# Food-Grade Pickering Emulsions Stabilized by Ultrasound-Treated Foxtail Millet Prolamin: Characterization and *In Vitro* Release Behavior of Curcumin

**DOI:** 10.3390/foods14030417

**Published:** 2025-01-27

**Authors:** Yu Guo, Yuewei Luo, Zhiyuan Ren, Xinpeng Zhang, Huiling Duan, Zhizong Liu, Xiaowen Wang

**Affiliations:** College of Food Science and Engineering, Shanxi Agricultural University, Taigu, Jinzhong 030801, China; a1466173328@163.com (Y.L.); 19581900333@163.com (Z.R.); m15254786422@163.com (X.Z.); duanhui_ling@126.com (H.D.); zhizongliu@163.com (Z.L.); wwxw11@163.com (X.W.)

**Keywords:** foxtail millet prolamin, ultrasound treatment, medium internal-phase pickering emulsions (MIPEs), rheological feature, curcumin

## Abstract

To date, extensive studies have focused on developing proteins as stabilizers to fabricate food-grade emulsions for encapsulating bioactive compounds aimed at targeted delivery. This paper aimed to develop a novel stabilizer using foxtail millet prolamin (FMP) to fabricate medium internal-phase Pickering emulsions (MIPEs) and investigate the stability and *in vitro* release behavior of curcumin (Cur) encapsulated within the MIPEs. Ultrasound treatment modified the secondary and tertiary structures of FMP, along with its particle size, zeta potential, and wettability, enhancing its functionality as a stabilizer for MIPEs. The MIPEs stabilized by 65% ultrasound-treated FMP (FMP-U) exhibited better rheological properties and stability, significantly improving the storage stability and antioxidant activity of Cur. *In vitro* digestion results demonstrated that the MIPEs delayed the release of Cur, achieving a final release rate of 84.0 ± 1.47% after 4 h of gastrointestinal digestion and the DPPH radical scavenging activity (RSA) of 39.9 ± 1.31%, which was notably higher than the RSA of free Cur in oil at only 5.8 ± 1.37%. Moreover, MIPEs with Cur increased the bioaccessibility of Cur. This study provides new insights into a novel delivery system designed with FMP-U for encapsulating hydrophobic compounds, thereby enhancing their stability, sustained release, and bioaccessibility.

## 1. Introduction

Pickering emulsions (PEs) are a type of emulsion stabilized by solid colloidal particles, characterized by being surfactant-free and highly stable, with strong resistance to coalescence and flocculation [1]. PEs have attracted significant scientific interest and have been widely utilized as effective delivery systems for encapsulating bioactive compounds, improving their stability by inhibiting degradation, oxidation, and reduction reactions, while also enhancing their bioavailability [2,3,4]. Biomacromolecules, especially polysaccharides and proteins, have attracted much attention as packing materials for developing PEs [5,6,7]. With the growing interest in vegetarian and vegan diets, the development of plant-based protein functionality has sparked significant research enthusiasm. Natural plant-based proteins, characterized by their poor aqueous solubility, ease of fabrication, biodegradability, and compatibility, hold substantial promise for developing food-grade PEs in the fields of food, agrochemicals, pharmaceutics, and personal care [2,5,8].

Among these plant-based particles, prolamin-based particles, such as gliadin, zein, and kafirin, have been extensively studied as effective stabilizers for the fabrication of food-grade PEs due to their tunable self-assembly behavior [8,9,10,11]. Prolamin, rich in glutamine and proline, is characterized by its resistance to digestive enzymes and low digestibility [9]. Therefore, PEs stabilized by prolamin-based particles have been developed as carriers for bioactive compounds, aiming to achieve controlled and prolonged release in the gastrointestinal tract [6,9,12,13,14]. Furthermore, PEs stabilized by prolamin-based particles have been shown to improve stability, bioaccessibility, and antioxidant activity [9,11,12]. So far, gliadin (isolated from wheat) and zein (from corn) have been relatively well-studied as stabilizers for the development of PEs. With the growing demand for encapsulation materials, the development of novel proteins as stabilizers for Pickering emulsions (PEs) has become a high priority.

Foxtail millet, one of the oldest crops, plays a crucial role in addressing global food security due to its drought tolerance and resistance to various environmental stresses [15,16]. It contains relatively high levels of protein (about 11.65%), including albumin, globulin, glutelin, and prolamin [17]. Prolamin in foxtail millet (FMP) is the major storage protein, accounting for 60–65% of the total protein, and is rich in hydrophobic amino acids such as proline, glutamic acid, valine, and phenylalanine [17]. Recent studies have shown that proline can interact with phytochemicals [18]. Therefore, FMP has the potential to effectively protect hydrophobic active compounds by adsorbing them into its hydrophobic core [12]. However, FMP exhibited lower emulsification properties due to its hydrophobic nature and high levels of disulfide bonding [17,19]. Various techniques have been applied to improve the emulsification properties of prolamin-based particles, including antisolvent, pH-modulation, and solvent evaporation methods, which all use chemical reagents. The use of chemical reagents contradicts principles of cleanliness and environmental protection. However, ultrasound has been demonstrated to be an effective, reliable, and environmentally-friendly technique for modifying and enhancing protein qualities [19]. It has been reported that ultrasound treatment effectively improves the emulsification properties of wheat gliadin, green wheat gliadin, and pea protein [20,21]. Additionally, studies have shown that ultrasound treatment can improve the functional attributes of FMP, including its solubility, foaming, and emulsifying properties [19,22]. Based on these findings, it is hypothesized that ultrasound-treated FMP (FMP-U) can serve as a stabilizer for forming PEs. To date, emulsions stabilized solely by FMP have not been reported.

Curcumin (Cur) is a hydrophobic polyphenol extracted from Curcuma longa plants. It has been reported that Cur offers various health benefits, such as antioxidation, anticancer, and anti-inflammatory [9,12]. However, its low water solubility, poor chemical stability, and limited bioavailability in the body hinder its widespread application. Encapsulating Cur in delivery systems is an effective strategy to address these issues [1,11,23].

The main objective of the present study was to develop oil-in-water Pickering emulsions stabilized by food-grade FMP-U particles for the encapsulation of Cur. The effect of ultrasound treatment on the physicochemical properties of FMP was systematically evaluated. Subsequently, the properties of PEs stabilized by FMP-U were analyzed. Additionally, the Cur-loaded PEs were prepared to evaluate their *in vitro* release behavior. This paper presents a novel and eco-friendly approach to fabricating food-grade PEs stabilized by modified FMP, serving as carriers for delivering fat-soluble bioactive compounds.

## 2. Materials and Methods

### 2.1. Materials and Chemicals

Foxtail millet (Jingu 21) was cultivated in Taigu District, Jinzhong City, Shanxi, China, and harvested in 2022. Fluorescent dyes (Nile Blue and Nile Red) were obtained from Sigma-Aldrich (Shanghai, China). Soybean oil was acquired from a local supermarket (Jinzhong, China). Cur (99%) was sourced from Shaanxi Taimei Bio-technology Co., Ltd. (Xi’an, Shaanxi, China). 1-anilino-8-naphthalene sulfonate (ANS) was obtained from Aladdin Biochemical Technology Co., Ltd. (Shanghai, China). Other reagents and solvents in the present study were of analytical grade.

### 2.2. Preparation of Foxtail Millet Prolamin

Foxtail millet was dehusked by a huller (JLGJ-45, Taizhou Grain Instrument C., Ltd., Taizhou, Zhejiang, China). The dehusked foxtail millet was milled using a pulverizer (YB-2000A, Yunbang Company, Wenzhou, Zhejiang, China) and passed through an 80-mesh sieve plate. The resulting foxtail millet powder was mixed with petroleum ether at a ratio of 1 g to 5 mL and stirred for 4 h at room temperature. The mixture was then centrifuged to remove the petroleum ether and air-dried overnight in a fume hood to eliminate any residual solvent. The mixture of the defatted foxtail millet powder and deionized water, at a weight-to-volume ratio of 1 g to 15 mL, was processed in a colloid mill (JMS60, Wuxi Maxwell Automation Technology Co., Ltd., Wuxi, Jiangsu, China) three times, with each milling session lasting 1 min. The pH value of the mixture was adjusted to 8.5–8.8 using 1 mol/L NaOH. The precipitate was resuspended in 70% ethanol at a weight-to-volume ratio of 1 g to 7 mL and stirred for 4 h at room temperature. The mixture was then centrifuged at 3500 rpm for 20 min and the supernatant was collected. Sodium chloride solution was added to the supernatant to achieve a sodium concentration of 0.3 g/100 g, and the mixture was stirred for 1 h at room temperature. Then the mixture was allowed to stand at 4 °C for 24 h to get the precipitate. The precipitate was washed with deionized water to remove excess sodium until the total dissolved solids (TDS) value approached that of deionized water. Then the isolated prolamin was freeze-dried and stored at −80 °C until use. The purity of the isolated foxtail millet prolamin (FMP) was 92%, as determined by the Kjeldahl method.

### 2.3. Ultrasound Treatment of Protein Solutions

FMP (5%, 25%, 45%, 55%, and 65%, *w*/*v*) were dispersed in deionized water and gently stirred at room temperature overnight for complete hydration. The FMP solutions were then subjected to ultrasound treatment in an ice bath using a probe ultrasonic processor (Scientz-IID, Ningbo Scientz Biotech., Ningbo, Zhejiang, China) with an output power of 420 W and an ultrasonic frequency of 20 kHz for 5 min. The ultrasonic processing was performed with a working cycle of 4 s on and 2 s off. The ultrasound-treated FMP (FMP-U) solutions were subsequently prepared and stored at 4 °C.

### 2.4. Characteristics of FMP and FMP-U Particles

#### 2.4.1. Particle Size Distribution (PSD) and ζ Potential

The particle size and ζ potential of FMP and FMP-U were determined using a Zeta-sizer Nano-ZS90 instrument (Malvern Instruments, Malvern, Worcestershire, UK).

#### 2.4.2. Wettability

The wettability of FMP and FMP-U was evaluated by measuring the three-phase contact angle (θ) using the sessile drop method [24]. The lyophilized FMP and FMP-U powders were compressed into films and soaked in soybean oil for 10 min. After removing excess oil from the film surfaces, a drop of deionized water (approximately 5 μL) was placed onto the surface of the films, and the contact surface was captured using an SDC-100 contact angle tester equipped with a high-speed camera (SINDIN Co., Ltd., Foshan, Guangdong, China). The Young equation was used to evaluate three-phase contact angle.

#### 2.4.3. Infrared (IR) Spectral Analysis

The IR spectra of FMP and FMP-U were determined using a Fourier transform infrared spectrophotometer (FTIR, Bruker, Ettlingen, Germany). Lyophilized powder was ground with KBr at the ratio of 1:20 and pressed into pellets for measurement. The spectra in the range of 4000 to 400 cm^−1^ were recorded with a resolution of 4 cm^−1^ and 16 scans. The spectra were corrected and analyzed using Peakfit software (Systat PeakFit 4.12, SeaSolve Software Inc., San Jose, CA, USA) to calculate the content of α-helix, β-sheet, β-turn, and random coil structures.

#### 2.4.4. UV Spectral and Fluorescence Analysis

The UV spectra of FMP and FMP-U aqueous solutions were measured using ultraviolet spectrophotometry (TU-1810, Beijing Persee General Instrument Co. Ltd., Beijing, China) across a wavelength range from 200 to 400 nm, with a scanning rate of 1.0 nm/s. The intrinsic and extrinsic fluorescence spectra of FMP and FMP-U aqueous solutions were analyzed using an F-7000 fluorescence spectrophotometer (AXimA-CFR, Hitachi, Tokyo, Japan). For intrinsic fluorescence analysis, an excitation wavelength of 280 nm was used, while an excitation wavelength of 390 nm was employed for extrinsic fluorescence analysis. For extrinsic fluorescence analysis, 4 mL of the sample solution was mixed with 20 μL ANS (8 mmol/L) and the resulting mixtures were analyzed.

#### 2.4.5. X-Ray Diffraction Technique (XRD)

The XRD patterns of FMP and FMP-U were performed with an X-ray diffractometer (D8 ADVANCE, Burker, Billerica, MA, USA). The sample was loaded on the stage, flattened and then placed in the sample holder. The diffraction angle (2*θ*) was taken between 5° and 65° with a step size of 5°/min. The emission voltage and current were 40 kV and 40 mA, respectively.

### 2.5. Emulsion Preparation

8 mL soybean oil was added to 8 mL FMP-U solution at different concentrations (5%, 25%, 45%, 55%, and 65%, *w*/*v*) and then the mixture was sheared at 20,000 rpm for 3 min by a homogenizer (FSH-2A, Yuexin Yiqi, Jiangsu, China) at room temperature [25]. Cur was thoroughly dissolved in soybean oil at a final concentration of 500 mg/L in the dark. Soybean oil with Cur was used in place of pure oil for preparing the emulsions. All the emulsions were stored at 4 °C and protected from light, for further analysis.

### 2.6. The Microstructure and Particle Size of Emulsions

The microstructure of the emulsions was analyzed using confocal laser scanning microscopy (CLSM) with a Leica DMRE-7 upright microscope equipped with a Leica TCS SP5 confocal laser scanning head (Leica Microsystems Inc, Heidelberg, Germany). Nile Red (excitation wavelength: 488 nm, emission wavelength: 498–640 nm) and Nile Blue (excitation wavelength: 638 nm, emission wavelength: 648–751 nm) were used as dyes to stain the emulsions. 40 μL 1 mg/mL Nile Red and 40 μL 1 mg/mL Nile Blue were gently mixed with 1 mL emulsions. After 30 min for equilibration, the dyed emulsions were observed by CLSM. The particle size distribution of emulsions was analyzed by a laser particle size analyzer (LS-POP (9), Zhuhai OMEC Instrument Co., Ltd., Zhuhai, Guangzhou, China).

### 2.7. Rheological Features

The rheological properties of the resulting emulsions were analyzed using a rheometer (MCR 301, Anton Paar, Physica, Graz, Austria) equipped with a parallel plate (60 mm diameter) at 1 mm gap [10]. Amplitude sweeps were conducted by applying a strain sweep ranging from 0.1% to 100% at a frequency of 1 Hz to determine the stress values within the linear viscoelastic region (LVR) for subsequent tests. Frequency sweeps were recorded between 1 to 100 Hz at the strain within LVR. The dynamic viscosity was measured over a shear rate range of 0.1–100 s^−1^ to evaluate the viscosity characteristics of the emulsions. Thixotropic recovery was assessed through alternating shear rate time sweeps with the shear rate set at 0.1 s^−1^ (30 s), 100 s^−1^ (30 s), and 0.1 s^−1^ (30 s). All tests were conducted at a constant temperature of 25 °C.

### 2.8. Stability of Emulsions

#### 2.8.1. Storage Stability

The emulsions were stored at 4 °C. To evaluate storage stability, the visual appearance of the emulsions was recorded at 7-day intervals, and the creaming index (CI) was calculated and monitored during 28 days of storage. The CI was calculated using the following Equation (1) [5]:(1)$$CI\left(\%\right)=H_{s}/H_{t}\times 100\%$$
where H_s_ means the height of the supernatant layer and H_t_ represents the total height of the emulsions.

#### 2.8.2. Thermal Stability

All emulsion samples were heated at 80 °C for 20 min and then cooled to room temperature [26]. The effect of thermal treatment on the stability of the emulsions was evaluated by recording visual appearance through photographs and measuring viscosity according to the method described in Section 2.7.

#### 2.8.3. Freeze–Thaw Stability

The samples were placed at −80 °C for 48 h and then thawed at room temperature. The visual appearance and viscosity were recorded and measured according to the method described in Section 2.7.

#### 2.8.4. pH Stability

The pH of the emulsions was adjusted to 3 and 11 using 1 mol/L HCl and NaOH, respectively [27]. Then, the appearance of the samples was recorded by taking photographs and the viscosity was measured using the method described in Section 2.7.

### 2.9. Storage Stability of Cur

The storage stability of free and encapsulated Cur was evaluated based on their antioxidant capacity, which was estimated by DPPH radical scavenging activity (RSA) [23]. 0.1 mmol/L DPPH-ethanol solution was added to the free and encapsulated Cur samples in a 1:1 ratio. The mixture of samples and ethanol in a 1:1 ratio served as the blank, and the control was the mixture of DPPH· solution and ethanol in a 1:1 ratio. All mixtures were vortexed vigorously and incubated for 30 min at room temperature in the dark. The supernatant was then collected to measure the absorbance at 517 nm. RSA was determined using the following Equation (2):(2)$$DPPH\;scavenging\;ration\;(\%)=(1-{(A}_{sample}-A_{blank})/A_{control})\times 100\%$$
where A_control_ is the absorbance of 2 mL DPPH with 2 mL ethanol; A_blank_ is 2 mL sample with 2 mL ethanol; A_sample_ is 2 mL sample with 2 mL DPPH.

### 2.10. Release Behavior of Cur Encapsulated in Emulsions Under in Vitro Digestion

#### 2.10.1. *In Vitro* Digestion

Oral phase: The emulsions loaded with Cur were mixed with simulated salivary fluid (SSF, pre-warmed to 37 °C) at a 1:1 ratio. After adjusting the pH to 6.8, the mixture was shaken for 3 min at 100 rpm.

Gastric phase: The oral-digested emulsions with Cur were mixed at a 1:1 ratio with simulated gastric fluid (SGF) containing 3.2 mg/mL pepsin (pre-warmed to 37 °C). The mixture was adjusted to pH 2.5 and shaken at 100 rpm for 2 h at 37 °C to simulate gastric digestion.

Intestine phase: The gastric-digested sample was immediately neutralized to pH 7.0 with 1 mol/L NaOH, then mixed with simulated intestinal fluid (SIF) (pre-warmed to 37 °C) containing 4.3 mg/mL bile salts and 14.8 mg/mL trypsin. The mixture was kept at 37 °C and continuously shaken at 120 rpm for 2 h. The pH was maintained at 7.0 with 0.1 mol/L NaOH, and the volume of NaOH added was recorded during 2 h intestinal digestion.

Gastrointestinal-digested samples were collected at 30-min intervals during digestion to evaluate the cumulative release rate of Cur, RSA, and the free fatty acids (FFAs) release rate.

#### 2.10.2. Cumulative Release Rate of Cur

The collected samples were centrifuged at 10,000 rpm for 10 min, and the supernatant was collected to measure the absorbance at 423 nm using a spectrophotometer. The amount of Cur was calculated and the cumulative release rate of Cur was determined by the following Formula (3):(3)$$Cumulative\;release\;rate\;(\%)=100\%\times C_{t}V_{t}/M$$
where C_t_ represents the concentration of Cur released into the digestive fluid at the corresponding time, V_t_ means the total volume of the digestive fluid at the corresponding time, and M means the total amount of Cur in the emulsion sample for digestion.

#### 2.10.3. The Scavenging Activity of DPPH‧

RSA of the released Cur was evaluated according to the method described in Section 2.9.

#### 2.10.4. Free Fatty Acids (FFAs) Release Rate

The release rate of FFAs was calculated using the following Equation (4):(4)$$FFAs\;release\;rate\;(\%)=V_{NaOH}\times 0.1\times M_{lipid}\times 100\%/{(2\times m}_{lipid})$$
where V_NaOH_ is the volume of NaOH added during intestinal digestion to maintain the pH value at 7.0; 0.1 is the molarity of NaOH used to keep the pH value at 7.0; M_lipid_ is the mean molecular weight of soybean oil (884 g/mol); m_lipid_ is the total weight of the soybean oil (g) initially added.

### 2.11. Statistical Analysis

The data were presented as the means ± standard deviations (SD) of three independent evaluations. SPSS 25 (SPSS Inc., Chicago, IL, USA) was employed to analyze the significant differences using an analysis of variance (ANOVA) with a Tukey post-hoc test at *p* < 0.05.

## 3. Results and Discussion

### 3.1. Properties of FMP Particles Treated by Ultrasound

SDS-PAGE was performed to detect the foxtail millet prolamin (FMP). Three protein bands, with molecular weights ranging from 10 kDa to 23 kDa, were clearly observed (Appendix A). This result is consistent with a previous report indicating that the prolamin fraction typically consists of three to four bands, with molecular weights ranging from 13 kDa to 27 kDa [17].

The physicochemical properties of proteins have been demonstrated to significantly influence their functional characteristics, particularly emulsifying properties [17]. Therefore, particle size distribution (PSD), zeta-potential, contact angle, infrared spectrum, UV spectra, exogenous fluorescence, and X-ray diffraction were employed to evaluate the effects of ultrasound treatment on the physicochemical characteristics of FMP. The PSD analysis revealed a reduction in the particle size of FMP following ultrasound treatment (Figure 1A). The protein aggregates are dissociated and fractured into small particles due to the disruption of the noncovalent interaction, such as van der Waals forces, caused by ultrasonic cavitation [17,22]. Similar findings have been reported for casein and rice bran protein [28,29]. The zeta potential values of the two samples were negative and ultrasound treatment significantly increased the absolute zeta potential value from −5.2 ± 1.17 mV to −8.7 ± 0.77 mV (Figure 1B), which was consistent with the findings reported by Jhan et al. and Sharma et al. [19,22]. Zeta potential reflects the surface charge of samples. The increase in the absolute value of zeta potential was primarily attributed to the enhanced exposure of charged groups caused by the degradation of protein aggregates induced by ultrasonication [19,30]. The contact angle, which reflects the hydrophilic and lipophilic properties of the samples, was lower in ultrasonically treated FMP (FMP-U, 98° ± 1.6°) compared to untreated FMP (109.5° ± 1.2°) (Figure 1C). This finding indicated that ultrasound treatment improves the hydrophilic properties of FMP, consistent with the results reported by Li et al. [19,31].

For the FTIR spectra of FMP and FMP-U, no new peaks appeared, nor did inherent peaks disappear (Figure 1D), consistent with the findings of Badar. et al. [32]. However, significant changes in secondary structure content were observed: an increase in β-sheet and random coil content, along with a decrease in α-helix and β-turn content. Similarly, Badar et al. also reported significant changes in the secondary structure of proteins subjected to ultrasound treatment [32]. The cavitation effect generated by ultrasound treatment disrupted the inter- or intra-molecular noncovalent bonds, leading to the denaturation and unfolding of proteins [17].

To further investigate the effect of ultrasound treatment on the structure of FMP, UV and fluorescence spectroscopy were employed to characterize the structures of FMP and FMP-U. The UV spectra showed an absorption peak for FMP at 280 nm, with no observed red or blue shift in the peak after ultrasound treatment (Appendix A). A second-derivative fitting based on UV spectra was conducted to analyze the microenvironmental changes in tyrosine and tryptophan residues. As shown in Figure 1E, the ratio of two peak-to-trough values marked as a and b (r = a/b) was calculated to estimate the conformational changes of tyrosine residues. The r values for FMP and FMP-U were 1.6 and 1.8, respectively. The increase in the r value indicated that the average polarity of tyrosine increased, while its hydrophobicity decreased [33]. The FTIR results demonstrated changes in the secondary structure due to ultrasound treatment, including the disruption of the folding and interactions of FMP, leading to the repositioning of tyrosine in a polar environment [17].

The maximum emission wavelengths of intrinsic and extrinsic fluorescence for FMP and FMP-U were 336 nm and 470 nm, respectively, consistent with the findings of Zhang et al. [33]. No shift in the maximum emission wavelength was observed for FMP after ultrasound treatment, but a decrease in fluorescence intensity was detected, indicating changes in the microenvironment of tryptophan. The reduction in intrinsic fluorescence intensity suggested that tryptophan residues shifted to a more hydrophilic region following ultrasound treatment. The ANS fluorescent probe method revealed differences in the tertiary structure, and the surface hydrophobicity of proteins was positively correlated with the fluorescence intensity of the ANS-protein complex [33]. A decrease in extrinsic fluorescence intensity was also observed. Ultrasound treatment induced the unfolding of FMP, exposing tryptophan residues from the core of the protein to a relatively stronger hydrophilic environment [34]. Consequently, ultrasound treatment resulted in the repositioning of tryptophan residues into a more hydrophilic environment.

The XRD measurements were employed to examine the crystalline structures of FMP and FMP-U (Figure 1H). Two characteristic diffraction angles (2*θ*) at 20° and 9° were observed in the XRD spectrum of FMP, corresponding to the β-sheet and α-helical structures of proteins, similar to the findings reported by Jhan et al. [22]. After ultrasound treatment, the intensities of these two peaks were reduced, reflecting the conformational changes in FMP induced by the cavitation process of ultrasonic waves [35]. A positive correlation between the diffraction intensity and the particle size has been demonstrated [36]. Therefore, the decrease in the diffraction intensity caused by ultrasound treatment indicated a reduction in particle size, which was consistent with the particle size distribution results.

Generally, ultrasound treatment altered the secondary and tertiary structures of FMP, resulting in changes in physicochemical properties and the repositioning of tyrosine and tryptophan residues into a more hydrophilic environment.

### 3.2. Appearance and Microstructure of the PEs Stabilized by FMP-U

Medium internal-phase Pickering emulsions (MIPEs) stabilized by FMP-U particles at different concentrations (5%, 25%, 45%, 55%, and 65%, *w*/*v*) with an oil fraction of 50% were prepared. The concentration of FMP-U influenced the visual appearance of MIPEs, and MIPEs stabilized by different concentrations of FMP-U exhibited varying behavior (Figure 2). The MIPEs stabilized by 5% and 25% FMP-U underwent rapid separation within 0.5–1 h into two layers, a creamed layer (top) and a water layer (bottom) (Figure 2A-1,B-1). This behavior can be attributed to the limited coalescence of the freshly formed oil droplets and the density difference between the two phases [1]. Oil droplets were clearly visible in MIPEs containing 0.5% FMP-U after 0.5–1 h of storage at room temperature, indicating an insufficient coverage of the oil–water interface by FMP-U particles (Figure 2A-1). The MIPEs stabilized by 45%, 55%, and 65% FMP-U adhered to the glass vials and exhibited gel-like behavior (Figure 2C-1–E-1).

The CLSM was employed to evaluate the microstructures of MIPEs stabilized by FMP-U, and the oil phase and FMP-U particles were stained with Nile red (green in Figure 2) and Nile blue (red in Figure 2), respectively. FMP-U (bright red fluorescence) formed an ordered interfacial barrier at the oil–water interface, encapsulating the oil spherical droplets (green fluorescence) to prevent the coalescence of oil droplets and stabilize the MIPEs. This observation confirmed that the emulsions were o/w emulsions (Figure 2) [1].

The developed emulsions exhibited a mono-modal particle size distribution ranging from 40 μm to 106 μm (Appendix A). As the concentration of FMP-U increased from 5% to 65%, the mean particle size of the emulsions significantly decreased from 85.58 μm to 66.66 μm, consistent with the findings of Lv et al. [1]. This reduction can be attributed to the increased availability of particles to effectively cover the oil–water interfacial area.

### 3.3. Rheological Behavior

The rheological properties of the resulting MIPEs are crucial for their application and processing design. Amplitude sweep, frequency sweep, viscosity, and thixotropy analysis were conducted to systematically evaluate the effect of FMP-U concentration on the rheological behavior of the MIPEs.

#### 3.3.1. Amplitude Sweep Analysis

The effects of strain magnitude on the properties of the MIPEs were shown in Figure 3A. Initially, the storage modulus (G′, representing elastic behavior) of the MIPEs was higher than the loss modulus (G″, representing viscous behavior), indicating the solid-like behavior of the MIPEs [7]. MIPEs stabilized by 65% FMP-U exhibited the highest elasticity with the highest storage modulus, while those stabilized by 5% FMP-U showed the lowest elasticity, indicating that the elasticity of the MIPEs increases with the concentration of FMP-U. The results suggested that the strength of the network structure formed by FMP-U was positively correlated with the particle concentration [7].

As the strain increased, leading to compression and deformation, the droplets moved into close proximity, increasing the forces between them [37]. Therefore, the linear viscoelastic region (LVR) was observed (Figure 3A). The length of LVR expanded with increasing FMP-U concentration, which can be attributed to the enhanced strength and integrity of the network structure formed at higher FMP-U concentrations, resulting in greater resistance to deformation [7]. Therefore, the MIPEs exhibited an enhanced capacity to withstand higher deformation with increasing FMP-U concentration.

As strain increased, a cross-over point between the storage modulus (G′) and the loss modulus (G″) was observed. Beyond this point, both G′ and G″ values decreased significantly, with G″ surpassing G′, suggesting a loss of elasticity due to the progressive disruption of the MIPEs structure at high strains [7,38]. Then, the system exhibited fluid-like behavior instead of pseudo-solid characteristics, which was attributed to the reorganization of emulsion droplets under high stress [39,40].

#### 3.3.2. The Oscillation Analysis

The dynamic oscillatory measurement was used to investigate the viscoelastic properties of MIPEs. The storage modulus (G′) higher than the loss modulus (G″) across the entire frequency range (1 to 100 Hz) suggested a typical gel-like rheological behavior [1,38]. The G′ and G″ values of the MIPEs with 45%, 55%, and 65% FMP-U exhibited independence on the applied frequency, further supporting their gel-like characteristics [7,41].

#### 3.3.3. The Flow Behavior Analysis

The viscosity (η) increased with rising particle concentrations, which was attributed to the enhanced interactions between droplets at higher FMP-U concentrations (Figure 3C, Table 1), consistent with the findings reported by Lv et al. [1]. The increase in viscosity effectively slowed droplet aggregation, thereby enhancing the stability of the MIPEs [7]. The viscosity of MIPEs stabilized by FMP-U was significantly higher than that of MIPEs stabilized by whey protein isolate at concentrations of 0.1–3.0% and by kafirin nanoparticles at concentrations of 0.25–2.0% [1,8]. The differences in the viscosity can be explained by the differences in the properties and the concentrations of stabilizers.

All the MIPEs exhibited an obvious reduction in apparent viscosity with increasing shear rate, indicating the shear-thinning behavior, which is a typical non-Newtonian behavior [1,7]. The increase in the shear rate disrupted the crosslinks between droplets, causing large clusters to break down into smaller ones [42].

#### 3.3.4. Thixotropy Analysis

A three-stage thixotropy analysis was conducted to evaluate the destruction and recovery of the structure of the MIPEs under an alternate cycle of low and high shear rates (0.1 and 100 s^−1^) (Figure 3D). In the initial stage, at a low shear rate, the viscosity of the samples remained relatively constant, indicating the steady state of their structure. A sharp decrease in the viscosity was observed in all samples when the shear rate increased from 0.1 s^−1^ to 100 s^−1^, suggesting the disruption of the network structures under a high shear rate. In the third stage, when the shear rate reduced from 100 s^−1^ back to 0.1 s^−1^, the viscosity of all samples increased immediately, indicating the recovery of the disrupted network structures. These findings demonstrated the thixotropic recovery ability of the MIPEs [10]. Differences in the thixotropic recovery rate were observed among MIPEs stabilized by varying FMP-U concentrations (Figure 3D, Table 1). The thixotropic recovery rate increased significantly from 60.23 ± 2.31% to 71.32 ± 0.55% as the FMP-U concentration increased from 5% to 65%. The results indicated that higher concentrations of FMP-U promoted the formation of a strengthened network structure between droplets, enhancing the thixotropic recovery ability of the resulting MIPEs following structural damage induced by mechanic force. This property is important for the application of MIPEs [7].

In general, increasing the FMP-U concentration enhanced the interactions between droplets, leading to distinct rheological behavior across different MIPEs. These variations open up opportunities for applications in the food industry, including alternatives of margarine, shortening, salad dressing, and potential use in 3D printing.

### 3.4. Stability Analysis

#### 3.4.1. Storage Stability

To estimate the effect of FMP-U concentration on the storage stability of the MIPEs, all samples were stored at 4 °C for 28 days. The obvious phase separation was observed for the MIPEs stabilized by 5% FMP-U within 0.5–1 h and in those with 25% FMP-U after 1 day of storage with a creaming index (CI) of 28% (Figure 4A,B). A visible water phase was observed in the MIPEs stabilized by 45% FMP-U after 3 days of storage, accompanied by a creaming index of 10%. Although no phase separation was observed in the MIPEs stabilized by 55% FMP-U after 7 days of storage, the sample exhibited flowability upon bottle inversion due to water layer separation. In contrast, the MIPEs stabilized by 65% FMP-U had no flowability and no significant changes in CI after 21 days, indicating a high storage stability. After 28 days of storage, the remarkable phase separation and water layer were observed in all samples. However, the CI at 28 d decreased progressively with increasing FMP-U concentration. At lower concentrations, the FMP-U particles were insufficient to stabilize the oil–water interface effectively. At higher FMP-U concentrations, the formation of a strengthened network structure between droplets in the system effectively inhibited creaming. The results indicated a negative correlation between the storage stability and FMP-U concentration, consistent with the results reported by Lv et al. [1].

Whey protein isolate fabricated by high hydrostatic pressure treatment.

#### 3.4.2. Heat and Freeze–Thaw Stability

Heat and freeze–thaw treatments are two of the common treatments in food processing. To assess the potential applications of the resulting MIPEs in the food industry, their responses to these treatments were evaluated in terms of appearance, CI, and flow behavior (Figure 4C–F). After heating at 80 °C for 20 min, phase separation and water leakage were observed in all samples, indicating that the MIPEs were sensitive to thermal treatment (Figure 4C). The reduction in emulsion stability can be attributed to the formation of protein aggregates induced by high-temperature processing [14]. The highest CI (55%) was observed in the MIPEs containing 5% FMP-U, while the lowest (12%) CI was found in those with 65% FMP-U, indicating a negative correlation between heat sensitivity and FMP-U concentration. The flow behavior analysis showed that all the emulsions subjected to heat treatment exhibited a shear-thinning behavior, consistent with the fresh MIPEs. Notably, the initial apparent shear viscosity of the MIPEs with 5% and 25% FMP-U increased compared to their fresh samples. However, a decrease in viscosity was observed in samples with higher FMP-U concentrations. This variation in viscosity can be explained by the alteration of FMP-U conformation due to heat treatment [17].

After freeze–thaw treatment, oil and water leakage were observed in all samples, suggesting disruption of the emulsions. Ice crystals formed during freezing might disrupt the structure of the MIPEs. Additionally, the freeze–thaw treatment may have disrupted the intra- or intermolecular interactions of proteins, leading to instability in the interfacial structure [26]. The decrease of CI for the freeze–thaw-treated MIPEs with the increase in FMP-U concentration suggested an improvement in freeze–thaw stability. The shear-thinning behavior was observed for all the emulsions after freeze–thaw treatment. Compared to the fresh samples, the initial viscosity of the samples increased, suggesting a progressive enhancement in the gel-like structures [1].

#### 3.4.3. pH Stability

The effect of pH conditions on the stability of the MIPEs was evaluated (Figure 4G–J). The MIPEs with 65% FMP-U exhibited stability at pH 11 without any oil or water leakage, whereas the other samples were found to be sensitive to pH changes (Figure 4G,H). Compared to pH 11, the samples exhibited greater sensitivity at pH 3, as evidenced by the higher CI across all samples at pH 3 (Figure 4H). The isoelectric point (IP) of foxtail millet prolamin is pH 5. pH 3 is closer to the IP of FMP, resulting in reduced surface charge and increased hydrophobicity, which promotes protein aggregation [17]. This may explain the observed decrease in initial viscosity compared to the fresh samples (Figure 4I). As a consequence, coalescence between oil droplets occurred due to the reduction in electrostatic repulsion [10]. At pH 11, the MIPEs showed an increase in viscosity at a shear rate of 0.1 s^−1^ (Figure 4J), possibly due to enhanced emulsion crosslinking during the pH increase process [27]. The decrease in CI with increasing FMP-U concentration across all emulsions demonstrated that higher FMP-U concentrations improved the pH stability of the emulsion. The flow behavior of all the MIPEs at both pH3 and pH11 showed shear-thinning behavior, which was similar to the fresh samples.

In general, the MIPEs stabilized by FMP-U exhibited sensitivity to heat, freeze–thaw, and pH conditions. Notably, the MIPEs containing 65% FMP-U showed excellent stability under storage, heat, freeze–thaw, and pH conditions. Consequently, its potential applications were further evaluated.

### 3.5. Application of MIPEs in Delivering Cur

Pickering emulsions show great potential for delivering oil-soluble nutrients [1,27]. Thus, the MIPEs with 65% FMP-U, which exhibited superior stability compared to the other formulations, were selected for Cur loading and further analysis.

#### 3.5.1. Appearance, Rheological Properties, and Protection of Cur

The particle size distribution of the Cur-loaded MIPEs with 65% FMP-U was similar to that of the original MIPEs (Appendix A). Furthermore, no flow was observed when the bottle was inverted, indicating high viscoelasticity and the preservation of the gel-like emulsion structure after Cur loading (Figure 5A). The effect of Cur on the rheological properties of the MIPEs stabilized by 65% FMP-U was evaluated using amplitude sweep, frequency sweep, viscosity, and thixotropy analysis (Figure 5B–E). The results demonstrated that the Cur-loaded MIPEs exhibited rheological properties similar to those of the MIPEs without Cur, including gel-like behavior with G′ exceeding G″, G′ and G″ remaining independent of the applied frequency, shear-thinning behavior, and excellent thixotropic recovery ability. A significant increase in the elasticity and a decrease in viscosity at a shear rate of 0.1 s^−1^ were observed after the MIPEs encapsulated with Cur. The phenolic hydroxyl groups of Cur interact with the carbonyl groups of amide bonds on proteins to form hydrogen bonds, inducing the configuration change of prolamin [12,43]. Therefore, the configuration transitions of prolamin induced changes in the interactions between prolamin molecules within the MIPEs, resulting in alterations in their viscoelastic properties.

The DPPH radical scavenging activity (RSA) of Cur was measured to evaluate the protection ability of the MIPEs during storage. Cur dissolved in soybean oil under the same storage condition was used as a control. The RSA of free Cur in oil was 45.3 ± 1.1%, whereas the RSA of Cur encapsulated in MIPEs was significantly higher at 85.6 ± 0.7%. This enhancement can be attributed to the ability of the MIPEs to improve the dispersion and solubility of Cur in the aqueous phase, facilitating more effective interactions with free radicals. Additionally, emulsification increases the number of reaction sites for interactions between Cur and free radicals. Similar findings have been reported by Peng et al. and Lei et al. [11,44]. Thus, the MIPEs significantly improved the bioavailability of Cur. During storage, the RSA of Cur decreased significantly due to oxidation and photodamage (Figure 5F) [1]. After 28 d, the RSA of free Cur decreased remarkably from 45.3 ± 1.1% to 18.6 ± 0.3%. A similar decrease was observed for Cur encapsulated in the MIPEs, with the RSA decreasing from 85.6 ± 0.7% to 63.9 ± 0.8%. However, the reduction rate in RSA of encapsulated Cur (25.38 ± 0.32%) was significantly smaller than that of free Cur (58.91 ± 1.62%), highlighting the superior protective ability of MIPEs. The network structure formed by FMP-U on the surface of oil droplets effectively protected Cur, enhancing its stability and prolonging its functional efficacy [1].

#### 3.5.2. *In Vitro* Simulated Digestion

The release of Cur encapsulated in the MIPEs was evaluated in a simulated gastrointestinal environment over increasing digestion time. During 120 min of gastric digestion, a rapid release Cur from the soybean oil was observed, attributed to the formation of smaller oil droplets [41]. The Cur release rate from the MIPEs reached 48.1 ± 2.36%, with a high RSA of 61.3 ± 1.31%, compared to free Cur in soybean oil (an RSA of 22.0 ± 1.49%) (Figure 6A,B). A similar release rate of Cur from delivery systems during gastric digestion was reported by Chen et al. and Chen et al. [41,43]. The previous results demonstrated that the MIPEs stabilized by FMP-U were sensitive to the low pH environment, which triggered the disruption of the emulsions structure to release Cur. Despite this, Cur in the MIPEs showed a higher RSA compared to free Cur in soybean oil, highlighting the protective role of FMP-U in the system. Following 120 min of intestinal digestion, the final Cur release rate from the MIPEs was 84.0 ± 1.47%, with an RSA of 39.9 ± 1.31%, significantly outperforming the RSA of free Cur in oil, which was only 5.8 ± 1.37%. A similar release rate of Cur was observed in gastric and intestinal digestion, and the release curve of Cur in gastrointestinal digestion was linear, which was consistent with the findings reported by Chen et al. [41]. Recent research reports that foxtail millet prolamin (FP)-based nanoparticles are an efficient delivery system for Cur, protecting it against degradation and enhancing its antioxidant activity [17]. A similar release behavior of Cur was reported [43]. However, chemical reagents were used in the fabrication of the nanoparticles, introducing potential risks to their application in the food industry. Ultrasound, on the other hand, is considered an environmentally-friendly technology. Meantime, the present study provided a simple way for fabricating the stable PEs. Thus, FMP-U modified by ultrasound has the potential for widespread application in the food industry. It has been demonstrated that FMP interacts with Cur through hydrophobic forces and hydrogen bonds, and the reduced digestibility of FMP-U is attributed to the higher content of β-sheet structures in FMP-U [12,17,18]. Thus, the FMP-U shell in the MIPEs can resist proteolytic enzymes, protecting the encapsulated Cur. Additionally, the interaction between FMP-U and Cur may offer protection by inhibiting oxidation. This may explain the sustained release of Cur with a higher activity.

The free fatty acid (FFA) release rate was measured to evaluate the degree of lipolysis. During intestinal digestion, free Cur in oil exhibited a higher FFA release rate, reaching a final value of 85.7 ± 0.56% (Figure 6C), consistent with the findings of Lei et al. [11]. In contrast, the FFA release rate from the MIPEs was slower, with a total FFA release of 68.6 ± 0.70%. The higher FFA release rate in free Cur in oil was due to its direct exposure to lipase. Conversely, the interfacial layers formed by FMP-U acted as a protective barrier, effectively reducing interactions between the digestive enzymes and the oil phase [11,41].

The bioaccessibility of Cur was evaluated following gastrointestinal digestion. Free Cur in oil showed a low bioaccessibility of 16.6 ± 0.78%, whereas Cur encapsulated in the MIPEs exhibited a significantly higher bioaccessibility of 65.7 ± 3.61%. During digestion, Cur undergoes degradation due to exposure to pH variations and digestive juices, resulting in poor bioavailability [11,27]. The higher bioaccessibility of Cur in the emulsions is attributed to the improved stability and solubility of Cur due to the increased surface area of the droplets [11,41]. Furthermore, Li et al. reported that the PEs stabilized by ultrasound-treated pea protein isolate/mung bean starch improved the bioaccessibility of β-carotene compared to those stabilized by untreated protein complexes [31]. The emulsifying properties of proteins can be improved by ultrasound treatment. This improvement contributes to the formation of a stable interfacial film, providing effective protection for the encapsulated components. Consequently, higher bioaccessibility of Cur was observed.

Therefore, the results demonstrated that FMP-U effectively controlled the release of Cur, enhanced its solubility and stability, and promoted its bioaccessibility.

## 4. Conclusions

In conclusion, ultrasound treatment modified the secondary and tertiary structures of FMP, along with physicochemical properties. The resulting FMP-U was demonstrated to be an excellent food-grade Pickering stabilizer for oil-in-water MIPEs. The concentration of FMP-U influenced the rheological behavior and the stability of the emulsions under storage, heat, freeze–thaw treatment, and pH changes. All the resulting MIPEs showed sensitivity to heat, freeze–thaw treatment, and pH variations. Among them, the MIPEs stabilized by 65% FMP-U exhibited a gel-like structure and were demonstrated to possess excellent storage stability. The MIPEs with 65% FMP-U provided effective protection for Cur and improved its bioaccessibility. In terms of *in vitro* digestion, the results clearly indicated that the MIPEs exhibited sustained release of Cur. These findings indicated that Pickering emulsions stabilized by 65% ultrasound-induced FMP was a promising delivery system for protection and sustained release of oil-soluble nutrients. The sensitivity of the MIPEs to heat, freeze–thaw treatment, and pH variations, as well as their application in real food, should be further studied to develop their potential food industrial applications.

## Figures and Tables

**Figure 1 foods-14-00417-f001:**
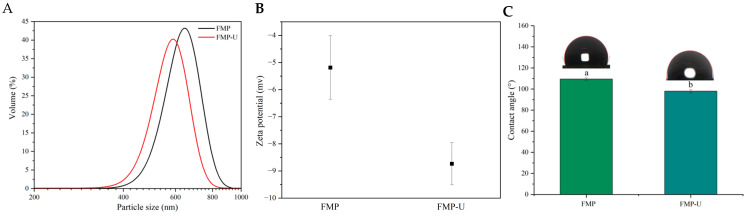
The characteristics of untreated (FMP) and ultrasound-treated (FMP-U) foxtail millet prolamin particles. (**A**): particle size; (**B**): zeta potential; (**C**): the three-phase contact angle (θ); (**D**): infrared spectrum; (**E**): second-order derivatives of UV spectra; (**F**): endogenous fluorescence; (**G**): exogenous fluorescence; (**H**): X-ray diffraction. The lowercase letters at the top of the columns and tables indicate significant differences between the samples at a significant level of *p* < 0.05. The purple dashed line (**E**) meas the peak-to-trough values.

**Figure 2 foods-14-00417-f002:**
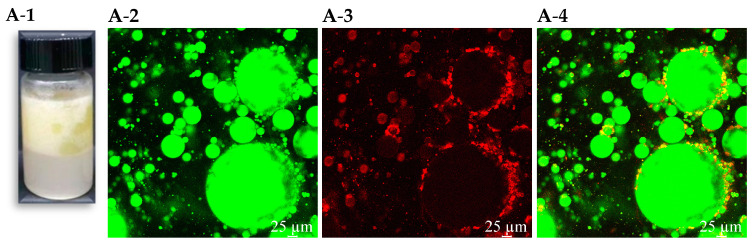
Visual appearance (**1**) and microstructure of Pickering emulsions stabilized by FMP-U observed by CLSM (oil core (**2**), protein shell (**3**), and emulsion structure (**4**)). (**A**): 5% FMP-U; (**B**): 25% FMP-U; (**C**): 45% FMP-U; (**D**): 55% FMP-U; (**E**) 65% FMP-U.

**Figure 3 foods-14-00417-f003:**
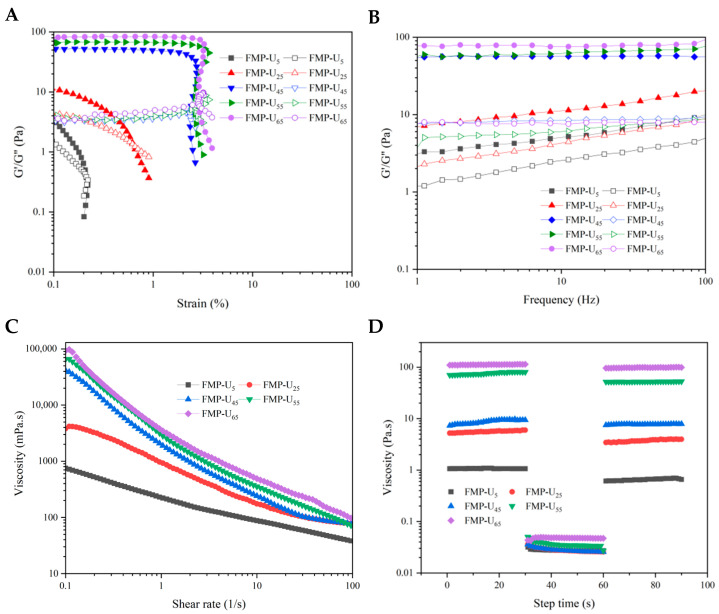
The rheological features of Pickering emulsions stabilized by FMP-U. (**A**): Amplitude sweep; (**B**): frequency sweep; (**C**): viscosity; (**D**): thixotropy.

**Figure 4 foods-14-00417-f004:**
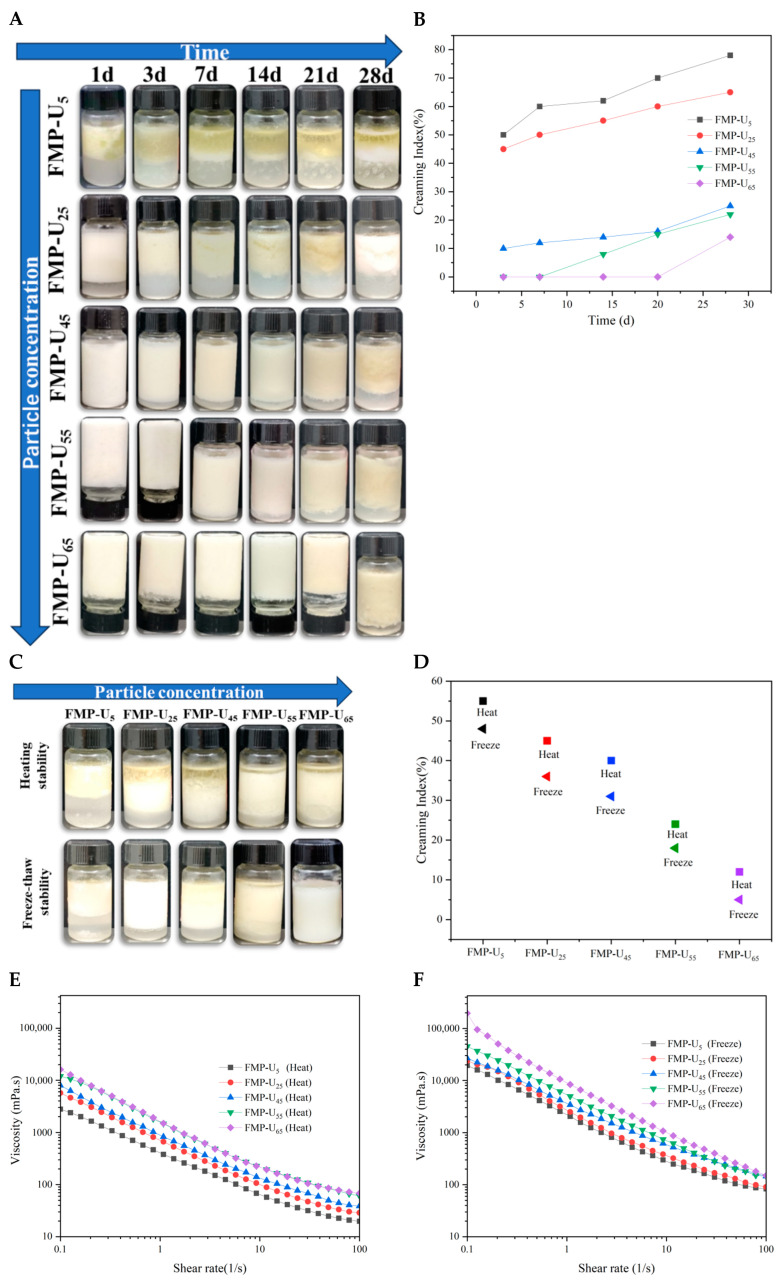
The stability of the Pickering emulsions. The appearance (**A**) and CI (**B**) of the MIPEs for storage stability; the appearance (**C**), CI (**D**), and flow behavior (**E**,**F**) of the MIPEs for heat and freeze–thaw stability; the appearance (**G**), CI (**H**) and flow behavior (**I**,**J**) of the MIPEs for pH stability.

**Figure 5 foods-14-00417-f005:**
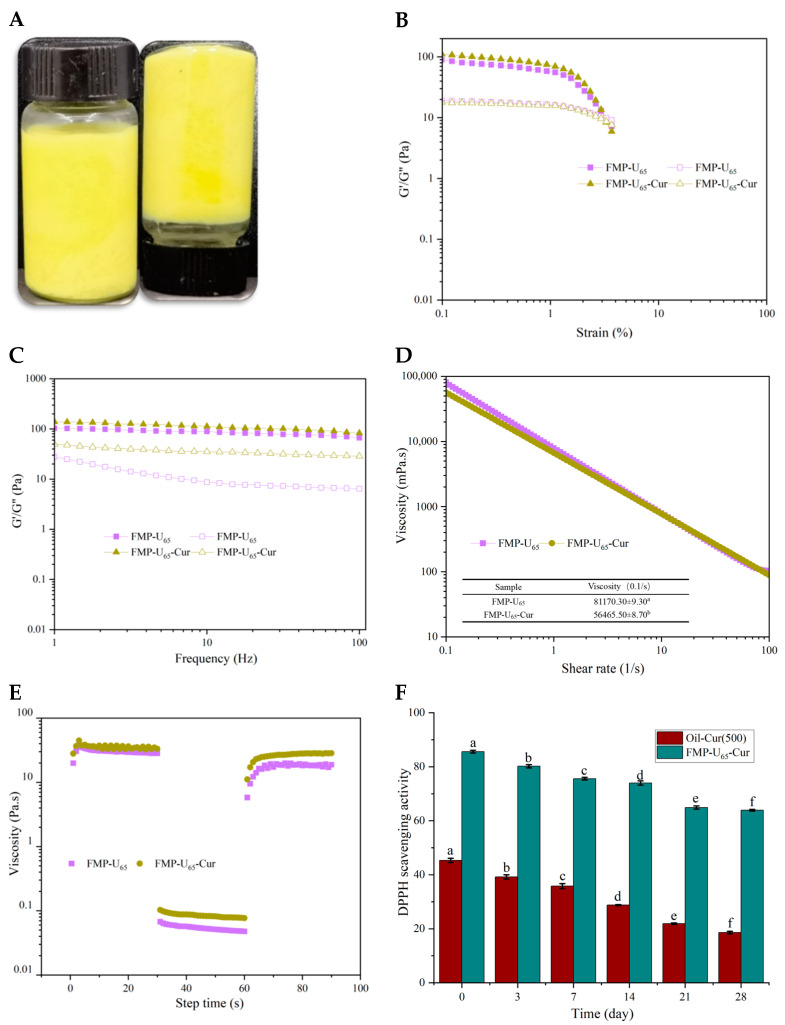
The appearance of the Pickering emulsions loaded with Cur (**A**); the rheological properties of emulsions: amplitude sweep (**B**), frequency sweep (**C**), viscosity (**D**), and thixotropy (**E**); the activity of Cur during storage (**F**). The different letters indicate significant differences between the samples at a significant level of *p* < 0.05.

**Figure 6 foods-14-00417-f006:**
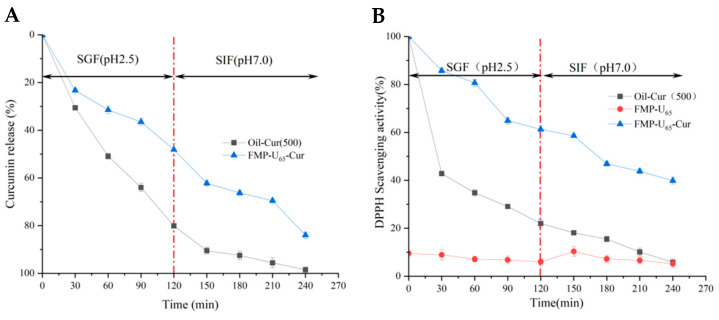
*In vitro* digestion of the MIPEs loaded with Cur. (**A**): Release of curcumin from the MIPEs; (**B**): digestion time dependence of RSA of curcumin; (**C**): digestion time dependence of %FFA release from the MIPEs; (**D**): bioaccessibility of curcumin loaded in the MIPEs. The lowercase letters at the top of the columns indicate significant differences between the samples at a significant level of *p* < 0.05.

**Table 1 foods-14-00417-t001:** Recovery rate of Pickering emulsions.

Sample	Recovery Rate(%)
FMP-U_5_	60.23 ± 2.31 ^a^*
FMP-U_25_	63.21 ± 1.11 ^b^
FMP-U_45_	65.42 ± 1.41 ^c^
FMP-U_55_	70.61 ± 0.25 ^d^
FMP-U_65_	71.32 ± 0.55 ^e^

* The lowercase letters indicate significant differences between the samples at a significant level of *p* < 0.05.

## Data Availability

The original contributions presented in this study are included in the article/Appendix A. Further inquiries can be directed to the corresponding author.

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
