# Peer review of "Food-Grade Pickering Emulsions Stabilized by Ultrasound-Treated Foxtail Millet Prolamin: Characterization and In Vitro Release Behavior of Curcumin"

_foods, 2025, doi:10.3390/foods14030417_

Round 1
Reviewer 1 Report
Comments and Suggestions for Authors
In this work, Guo et al., describes the an eco-friendly approach for fabricating food-grade Pickering emulsion stabilized by ultrasound-modified Foxtail millet prolamin. In addition, the authors demonstrated the application of the prepared medium internal-phase Pickering emulsion in enhancing the stability and bio-accessibility of curcumin, as a model hydrophobic bioactive compound in foods. This is a timely and relevant work given the increasing global focus in the application of plant-derived proteins as sustainable and healthy alternatives in food products. The work is technically sound and is well-presented. The arguments made are cogent and supported by strong evidence. In all, it is a worthy research and will make a significant contribution in foods. That said, a number of minor issues require prompt attention by the authors. These have been noted in the points below.
-Title: Authors should be consistent in the format of the title. ‘In vitro’ should all be in lower case where the first letter in ‘behavior’ should be in upper case.
-Line 22: achieving a final release rate of 84.0% ± 1.47 … Please include the duration.
Lines 36-37: Can the authors mention some of the biomacromolecules used as package agents for stabilizing PE, and also indicate the advantages of using proteins, especially plant-based proteins as opposed to their counterpart biomacromolecules?
-Lines 37-39: “Among these, natural food proteins, characterized by their low cost, ease of fabrication, biodegradability, and compatibility, hold substantial promise for developing food-grade Pes…” Authors should be more specific in their assertions. For instance, animal-based food proteins are neither cheap nor easily purified. The afore-highlighted sentence should thus be corrected accordingly.
-Section 2.2. Preparation of foxtail millet prolamin. Some details have been omitted in the description of this method. For example, how was foxtail millet dehusked and how was the dehusked foxtail millet pulverized. Authors should provide complete step-by-step detail in every method used as well as full details on the instruments to allow for reproducibility/replicability of the described method.
-Lines 160-161: “soybean oil 160 with 500 mg/mL Cur was used in place of pure oil for preparing the emulsions loaded with Cur.” Please clarify how the 500 mg/mL curcumin was prepared.
-Line 169: dyed emulsions… Step-by-step full details on how the dyed emulsions were prepared should be provided.
-Lines 252-254: Indicate how multiple comparisons was performed as well as the company, city country that provided that owned the statistical software.
-Authors should recheck Figure 1E. The horizontal axis is in nm and the range is from 250-315. Please verify this Figure and also explain in detail the actual changes in secondary structural components induced by ultrasound treatment of FMP.
-Lines 310-312: “The decrease in extrinsic fluorescence intensity demonstrated that the ultrasound treatment caused FMP to expose more hydrophilic groups while folding some hydrophobic groups, thereby reducing its hydrophobicity [34].” Some readers would find this as quite speculative. Could the authors explain in more detail how this occurs and the actual ‘groups’ that might be involved?
-Lines 316-318: Authors should indicate what the ‘characteristic (2θ) at 20Ëš and 9Ëš’ represent.
-The comparative analysis with other plant proteins used in MIPE could be expanded further on in the Results and discussions sections to provide better context.
-What are the main limitations of using FMP-U as Pickering stabilizer for the formation of medium internal-phase Pickering emulsions, and what are the potential challenges in the real-world and widespread application of FMP-U stabilized MIPE for delivery of hydrophobic bioactive compounds in food product?
-There are many incidences of poor usage of English language as well as syntax, grammar and typographical errors. A thorough editing of English language will greatly benefit the manuscript.
-The authors should also strive for improving the originality of the writeup by reducing Similarity Index with previously published materials in the literature.
Thank you.
Author Response
Dear reviewers,
Thanks very much for the careful review of our manuscript. We have considered the comments carefully and revised the manuscript according to your comments and suggestions. A revised manuscript with the correction sections marked in red for easy checking/editing purposes was attached. We deeply appreciate your consideration of our manuscript. The point-to-point responses have been made and listed as follows.
Thank you very much for your help. I am looking forward to hearing from you soon.
With kindest regards,
Yours sincerely,
Yu Guo
Comment 1: Title: Authors should be consistent in the format of the title. ‘In vitro’ should all be in lower case where the first letter in ‘behavior’ should be in upper case.
Reply: Thank you for your suggestion. We revised the title.
Line 2-4: Food-Grade Pickering Emulsions Stabilized by Ultrasound-Treated Foxtail Millet Prolamin: Characterization and in vitro Release Behavior of Curcumin
Comment 2: Line 22: achieving a final release rate of 84.0% ± 1.47 … Please include the duration.
Reply: Thank you for your suggestion. We have provided the details.
Line 21-25: In vitro digestion results demonstrated that the MIPEs delayed the release of Cur, achieving a final release rate of 84.0% ± 1.47 after 4 hours of gastrointestinal digestion and the DPPH radical scavenging activity (RSA) of 39.9% ± 1.31, which was notably higher than the RSA of free Cur in oil at only 5.8% ± 1.37%.
Comment 3: Lines 36-37: Can the authors mention some of the biomacromolecules used as package agents for stabilizing PE, and also indicate the advantages of using proteins, especially plant-based proteins as opposed to their counterpart biomacromolecules?
Reply: Thank you for your suggestion. We revised this part. The biomacromolecules used as package agents for stabilizing PE include polysaccarides and proteins and we listed these compounds in the Manuscript (Line 37-44).
Line 37-44: Biomacromolecules, especially polysaccharides and proteins, have attracted much attention as packing materials for developing PEs [5-7]. With the growing interest in vegetarian and vegan diets, the development of plant-based protein functionality has sparked significant research enthusiasm. Natural plant-based proteins, characterized by their poor aqueous solubility, ease of fabrication, biodegradability, and compatibility, hold substantial promise for developing food-grade PEs in the fields of food, agrochemicals, pharmaceutics, and personal care[2,5,8].
Comment 4: Lines 37-39: “Among these, natural food proteins, characterized by their low cost, ease of fabrication, biodegradability, and compatibility, hold substantial promise for developing food-grade Pes…” Authors should be more specific in their assertions. For instance, animal-based food proteins are neither cheap nor easily purified. The afore-highlighted sentence should thus be corrected accordingly.
Reply: Thank you for your suggestion. We revised this part (Line 39-44).
Line 39-44: With the growing interest in vegetarian and vegan diets, the development of plant-based protein functionality has sparked significant research enthusiasm. Natural plant-based proteins, characterized by their poor aqueous solubility, ease of fabrication, biodegradability, and compatibility, hold substantial promise for developing food-grade PEs in the fields of food, agrochemicals, pharmaceutics, and personal care[2,5,8].
Comment 5: Section 2.2. Preparation of foxtail millet prolamin. Some details have been omitted in the description of this method. For example, how was foxtail millet dehusked and how was the dehusked foxtail millet pulverized. Authors should provide complete step-by-step detail in every method used as well as full details on the instruments to allow for reproducibility/replicability of the described method.
Reply: Thank you for your suggestion. We have added the details of the dehusker and pulverizer. Foxtail millet was first placed into the rice huller for dehusking. Then the dehusked millet was placed into the pulverizer. These two instruments do not require any parameters to be set.
Lines 103-105: Foxtail millet was dehusked by a huller (JLGJ-45, Taizhou Grain Instrument C., Ltd., Zhejiang, China). The dehusked foxtail millet was milled using a pulverizer (YB-2000A, Yunbang Company, Zhejiang, China) and passed through an 80-mesh sieve plate.
Comment 6: Lines 160-161: “soybean oil 160 with 500 mg/mL Cur was used in place of pure oil for preparing the emulsions loaded with Cur.” Please clarify how the 500 mg/mL curcumin was prepared.
Reply: Thank you for your suggestion. We corrected the final concentration of Cur in soybean oil as 500mg/L. Also this part was revised.
Line: 172-175: Cur was thoroughly dissolved in soybean oil at a final concentration of 500 mg/L in the dark. Soybean oil with Cur was used in place of pure oil for preparing the emulsions. All the emulsions were stored at 4 °C, protected from light, for further analysis.
Comment 7: Line 169: dyed emulsions… Step-by-step full details on how the dyed emulsions were prepared should be provided
Reply: Thank you for your suggestion. We revised it and added the details.
Line 182-183: 40μL 1mg/mL Nile Red and 40μL 1mg/mL Nile Blue were gently mixed with 1mL emulsions.
Comment 8: Lines 252-254: Indicate how multiple comparisons was performed as well as the company, city country that provided that owned the statistical software.
Reply:Thank you for your suggestion. The details for the statistical analysis were added in the manuscript.
Line 268-270: SPSS 25 (SPSS Inc., Chicago, Illinois, USA) was employed to analyze the significant differences using an analysis of variance (ANOVA) with with a Tukey post-hoc test at p < 0.05.
Comment 9: Authors should recheck Figure 1E. The horizontal axis is in nm and the range is from 250-315. Please verify this Figure and also explain in detail the actual changes in secondary structural components induced by ultrasound treatment of FMP.
Reply: Thank you for your suggestion. We redrew the Figure 1E and the revised Figure has been updated. The changes in secondary structure of FMP induced by ultrasound treatment were also listed in the manuscript (Line 305-309, 319-322)
Line 305-309: Similarly, Badar et al. also reported significant changes in the secondary structure of proteins subjected to ultrasound treatment[32]. The cavitation effect generated by ultrasound treatment disrupted the inter- or intra-molecular noncovalent bonds, leading to the denaturation and unfolding of proteins[17].
Line: 319-322: The FTIR results demonstrated changes in the secondary structure due to ultrasound treatment, including the disruption of the folding and interactions of FMP, leading to the repositioning of tyrosine in a polar environment[17].
Comment 10: Lines 310-312: “The decrease in extrinsic fluorescence intensity demonstrated that the ultrasound treatment caused FMP to expose more hydrophilic groups while folding some hydrophobic groups, thereby reducing its hydrophobicity [34].” Some readers would find this as quite speculative. Could the authors explain in more detail how this occurs and the actual ‘groups’ that might be involved?
Reply: Thank you for your suggestion. According to published papers, the decrease of extrinsic fluorescence intensity can be explained by the repositioning of tryptophan residues into a polar environment. We have revised these sentences accordingly.
Line 331-334: A decrease in extrinsic fluorescence intensity was also observed. Ultrasound treatment induced the unfolding of FMP, exposing tryptophan residues from the core of the protein to a relatively stronger hydrophilic environment[34].
Comment 11: Lines 316-318: Authors should indicate what the ‘characteristic (2θ) at 20Ëš and 9Ëš’ represent.
Reply: Thanks for your suggestions. The ‘characteristic (2θ) at 20Ëš and 9Ëš’ represent the the β-sheet and α-helical structures of proteins. The details were added in the manuscript.
Line 338-340: Two characteristic diffraction angles (2θ) at 20Ëš and 9Ëš were observed in the XRD spectrum of FMP, correpresenting the β-sheet and α-helical structures of proteins, similar to the findings reported by Jhan et al.[22].
Comment 12: The comparative analysis with other plant proteins used in MIPE could be expanded further on in the Results and discussions sections to provide better context.
Reply: Thanks for your suggestions. The manuscript was revised and the discussion between MIPEs stabilized by FMP-U and MIPEs stabilized by other protein particles were added.
Line 421-425: The viscosity of MIPEs stabilized by FMP-U was significantly higher than that of MIPEs stabilized by whey protein isolate at concentrations of 0.1% - 3.0% and by kafirin nanoparticles at concentrations of 0.25% - 2.0%[1, 8]. The differences in the viscosity can be explained by the differences in the properties and the concentrations of stabilizers.
Comment 13: What are the main limitations of using FMP-U as Pickering stabilizer for the formation of medium internal-phase Pickering emulsions, and what are the potential challenges in the real-world and widespread application of FMP-U stabilized MIPE for delivery of hydrophobic bioactive compounds in food product?
Reply: Thanks for your suggestions. The present study only presented the properties of the MIPEs stabilized by FMP-U analyzed without of food matrix. The further research should be conducted on the influence of the real food matrix on the properties of the MIPEs, especially their stability, as well as the effect of the MIPEs on the texture and flavor properties of food.
Line 646-648: The sensitivity of the MIPEs to heat, freeze-thaw treatment, and pH variations, as well as their application in the real food, should be further studied to develop the potential food industrial applications.
Comment 14: There are many incidences of poor usage of English language as well as syntax, grammar and typographical errors. A thorough editing of English language will greatly benefit the manuscript.
Reply: Thanks for your suggestions. We revised the manuscript and highlighted the changes in red.
Comment 15: The authors should also strive for improving the originality of the writeup by reducing Similarity Index with previously published materials in the literature.
Reply: Thanks for your suggestions. We revised the manuscript and highlighted the changes in red.
Reviewer 2 Report
Comments and Suggestions for Authors
Recommendation: Reconsider after major revisions.
The study explores the development of food-grade Pickering emulsions stabilized by ultrasound-treated foxtail millet prolamin (FMP-U) to encapsulate curcumin. It demonstrates how ultrasound modification enhances FMP's emulsifying properties, leading to emulsions with improved stability, rheological behavior, and bioaccessibility of curcumin. The findings highlight the potential of FMP-U as a sustainable and efficient delivery system for hydrophobic bioactive compounds in the food industry. Below are detailed suggestions for significant improvement.
Major comments:
1. The objectives are clear, but they can be strengthened by explicitly outlining the novelty of the approach compared to existing research on prolamin-based Pickering emulsions. Highlighting gaps addressed by ultrasound-treated FMP would further emphasize the study's significance.
2. It would be beneficial to include untreated FMP-stabilized emulsions as a baseline comparison throughout all experiments, particularly for stability tests.
3. The rheological tests are comprehensive, but the discussion on practical implications (e.g., potential applications in food texture design) can be expanded. For instance, link the gel-like behavior of emulsions stabilized by 65% FMP-U to potential industrial applications such as spreads or dressings.
4. The freeze-thaw and thermal stability results highlight the sensitivity of the MIPEs. Discussing potential additives or modifications to improve these properties could add value.
5. The results on Cur bioaccessibility and release behavior are promising. However, discussing potential interactions between FMP and Cur during digestion, and how these might affect absorption and efficacy, would provide deeper insight.
6. Include comparisons with alternative encapsulation methods to contextualize the efficacy of the FMP-U-based system.
7. While the manuscript references relevant studies, a more detailed comparison with similar protein-stabilized emulsions, especially those involving ultrasound modifications, would strengthen the discussion.
8. The manuscript is generally well-written, but minor grammatical errors and redundancies exist. For example, simplify overly technical sentences in the abstract and conclusion for broader accessibility.
Ensure uniform terminology throughout (e.g., "FMP-U" vs. "ultrasound-treated FMP").
Comments on the Quality of English Language
Quality of english language is moderate.
Author Response
[1]Dear Editor and Reviewer,
Thanks very much for the careful review of our manuscript. We have considered the comments carefully and revised the manuscript according to your comments and suggestions. A revised manuscript with the correction sections marked in red for easy checking/editing purposes was attached. We deeply appreciate your consideration of our manuscript. The point-to-point responses have been made and listed as follows.
Thank you very much for your help. Looking forward to hearing from you soon.
With kindest regards,
Yours sincerely,
Yu Guo
Reviewer 2
Comment 1: The objectives are clear, but they can be strengthened by explicitly outlining the novelty of the approach compared to existing research on prolamin-based Pickering emulsions. Highlighting gaps addressed by ultrasound-treated FMP would further emphasize the study's significance.
Reply: Thanks for your suggestions. We revised the manuscript and listed more details.
Line 67-72: Various techniques have been applied to improve the emulsification properties of prolamin-based particles, including antisolvent, pH-modulation, and solvent evaporation methods, which all use chemical reagents. The use of chemical reagents contradicts principles of cleanliness and environmental protection. However, ultrasound has been demonstrated to be an effective, reliable, and environment-friendly technique for modifying and enhancing protein qualities[19].
Comment 2: It would be beneficial to include untreated FMP-stabilized emulsions as a baseline comparison throughout all experiments, particularly for stability tests.
Reply: Thanks for your suggestions. We tested different concentrations of FMP (5%-65%) to stabilized MIPEs and also HIPEs, but the resulting emulsions were unstable, and the phase separation was observed in short time. Based on the preliminary experiment, we explored different techniques to modify FMP to improve the emulsification property. We found ultrasound treatment could improve the stable of PEs, especially the storage stability. Thus, we only provided the results of the MIPEs stabilized by FMP-U.
Comment 3: The rheological tests are comprehensive, but the discussion on practical implications (e.g., potential applications in food texture design) can be expanded. For instance, link the gel-like behavior of emulsions stabilized by 65% FMP-U to potential industrial applications such as spreads or dressings.
Reply: Thanks for your suggestions. We revised the manuscript.
Line 421-425: The viscosity of MIPEs stabilized by FMP-U was significantly higher than that of MIPEs stabilized by whey protein isolate at concentrations of 0.1% - 3.0% and by kafirin nanoparticles at concentrations of 0.25% - 2.0%[1, 8]. The differences in the viscosity can be explained by the differences in the properties and the concentrations of stabilizers.
Comment 4: The freeze-thaw and thermal stability results highlight the sensitivity of the MIPEs. Discussing potential additives or modifications to improve these properties could add value.
Reply: Thanks for your suggestions. We noticed the sensitivity of the resulting MIPEs to heat, freeze-thaw treatment, and pH variations. The improvement of the stability of the MIPEs was observed through adding other components. These results and conclusions will be presented on our next paper that we are trying to publish.
Comment 5: The results on Cur bioaccessibility and release behavior are promising. However, discussing potential interactions between FMP and Cur during digestion, and how these might affect absorption and efficacy, would provide deeper insight.
Reply: Thanks for your suggestions. Based on the published papers, we discussed the reason for sustained release of Cur with a higher RSA.
Line 596-602: It has been demonstrated that FMP interacts with Cur through hydrophobic forces and hydrogen bonds, and the reduced digestibility of FMP-U is attributed to the higher content of β-sheet structures in FMP-U[12, 17, 18]. Thus, the FMP-U shell in the MIPEs can resist proteolytic enzymes, protecting the encapsulated Cur. Additionally, the interaction between FMP-U and Cur may offer protection by inhibiting oxidation. This may explain the sustained release of Cur with a higher activity.
Comment 6: Include comparisons with alternative encapsulation methods to contextualize the efficacy of the FMP-U-based system.
Reply: Thanks for your suggestions. We revised the manuscript and discussed the MIPEs and foxtail millet prolamin (FP)-based nanoparticles for delivering activity compounds.
Line 590-598: Recent research reports that foxtail millet prolamin (FP)-based nanoparticles are an efficient delivery system for Cur, protecting it against degradation and enhancing its antioxidant activity[17]. A similar release behavior of Cur was reported[43]. However, chemical reagents were used in the fabrication of the nanoparticles, introducing potential risks to their application in the food industry. Ultrasound, on the other hand, is considered an environment-friendly technology. Meantime, the present study provided a simple way for fabricationg the stable PEs. Thus, FMP-U modified by ultrasound has the potential for widespread application in the food industry.
Comment 7: While the manuscript references relevant studies, a more detailed comparison with similar protein-stabilized emulsions, especially those involving ultrasound modifications, would strengthen the discussion.
Reply: Thanks for your suggestions.
Line 624-6308: Furthermore, Li et al. reported that the PEs stabilized by ultrasound-treated pea protein isolate/mung bean starch improved the bioaccessibility of β-carotene compared to those stabilized by untreated protein complexes[31]. The emulsifying properties of proteins can be improved by ultrasound treatment. This improvement contributes to the formation of a stable interfacial film, providing effective protection for the encapsulated components. Consequently, higher bioaccessibility of Cur was observed.
Comment 8: The manuscript is generally well-written, but minor grammatical errors and redundancies exist. For example, simplify overly technical sentences in the abstract and conclusion for broader accessibility. Ensure uniform terminology throughout (e.g., "FMP-U" vs. "ultrasound-treated FMP").
Reply: Thanks for your suggestions. We revised the manuscript and the corrected sections are highlighted in red.
Round 2
Reviewer 2 Report
Comments and Suggestions for Authors
All my suggestions has been addressed by the authors. I recommend to publish in its present form.
Comments on the Quality of English LanguageQuality of english language is good.